# Analysis of Runoff Changes in the Wei River Basin, China: Confronting Climate Change and Human Activities

Ruirui Xu [1,2,3], Chaojun Gu [4], Dexun Qiu [1,2,3], Changxue Wu [2], Xingmin Mu [1,2,3,*] and Peng Gao [1,2,3,*]

1  State Key Laboratory of Soil Erosion and Dryland Farming on the Loess Plateau,
   Institute of Soil and Water Conservation, Chinese Academy of Sciences and Ministry of Water Resources,
   Xianyang 712100, China
2  Institute of Soil and Water Conservation, Northwest A & F University, Xianyang 712100, China
3  University of Chinese Academy of Sciences, Beijing 100049, China
4  Yangtze River Basin Monitoring Center Station for Soil and Water Conservation,
   Changjiang Water Resources Commission, Wuhan 430012, China
*  Correspondence: xmmu@ms.iswc.ac.cn (X.M.); gaopeng@ms.iswc.ac.cn (P.G.)

**Abstract:** Abrupt runoff reduction in the Wei River Basin (WRB) has attracted extensive attention owing to climate change and human activities. Nevertheless, previous studies have inadequately assessed the respective contributions of climate variability and human activities to runoff change on different spatial scales. Using Mann–Kendall and Pettitt's methods, this study identified long-term (1970–2018) changes in hydro-meteorological variables. Furthermore, the Budyko-based method was used to quantify the influence of climate change and human activities on runoff change at different spatial scales of the WRB, including the whole WRB, three sub-basins, and sixteen catchments. The results show that a significant decrease trend was identified in runoff at different spatial scales within the WRB. Runoff in almost all catchments showed a significant downward trend. Temperature, potential evapotranspiration, and the parameter n showed significant increases, whereas no significant trend in precipitation was observed. The change in runoff was mainly concentrated in the mid-1990s and early 2000s. Anthropogenic activities produced a larger impact on runoff decrease in the WRB (62.8%), three sub-basins (53.9% to 65.8%), and most catchments (–47.0% to 147.3%) than climate change. Dramatic catchment characteristic changes caused by large-scale human activities were the predominant reason of runoff reduction in the WRB. Our findings provide a comprehensive understanding of the dominate factors causing runoff change and contribute to water resource management and ecosystem health conservation in the WRB.

**Keywords:** runoff; climatic variation; land cover change; Budyko framework; Wei River Basin





## 1. Introduction

Runoff, a key component of the hydrological cycle, is strongly influenced by climate variability and anthropogenic activities, and therefore, changes dramatically at various spatial and temporal scales [1,2]. Climate change includes the changes in precipitation, temperature, and potential evaporation, which largely affect the future runoff of a basin [3]. In addition to climate change, anthropogenic factors such as ecological restoration programs, urbanization, industrialization, reservoir operation, large-scale irrigation, and drainage have strongly shaped the runoff regime [4]. Previous studies have shown that global runoff has increased over the 20th century [5]. However, river flow has decreased significantly in many basins worldwide, such as the Spanish basins [6], the Lake Issyk-Kul Basin [7], East African basins [8], as well as some basins in China, such as the Yellow River Basin [9] and the Shiyang River Basin [10], especially in arid and semi-arid regions [11,12]. The continuing decline in runoff has caused a severe water crisis and ecosystem imbalance, threatening the high state and sustainable development of the economy and society [13].

According to the IPCC report [14], the global climate has been undergoing extensive and complicated change in the past century, which aggravates the uneven distribution of precipitation (P) [15,16] and potential evapotranspiration (PET) [17], and consequently influences water balance. For rivers in tropical areas at low latitudes, the long-term tendency of runoff change is generally consistent with that of P [18,19]; however, for rivers in middle latitudes and some alpine regions, the correlation between runoff and P is weak [20,21]. The direct or indirect interference of runoff by human activities, including deforestation, afforestation, reservoir operation, large-scale irrigation, and urbanization, is an important reason for this regional difference [9,22,23]. Quantitative assessment of the impacts of climate variability and anthropogenic activities on runoff regimes at the regional scale is therefore vital to water resource management.

Several methods have been applied to analyze the influence of climate variability and human intervention on alterations in runoff [24–26]. Empirical statistical methods can achieve results quickly and easily; however, it is difficult to interpret complex and non-linear hydrological behaviors. Hydrological models, although powerful, can be constrained by the inherent variability and uncertainty in the impact of human activities on hydrological systems [27], and the difficulty in obtaining numerous input data related to human activities, such as water abstraction and irrigation data [28,29]. With the Budyko [30] hypothesis, various Budyko-based approaches have been widely used to explore the non-linear interactions among climate, landscape, and hydrology due to its straightforward calculations and physical mechanisms [31]. It suggests that the ratio of precipitation to potential evapotranspiration (P/PET) is the primary factor determining the hydrological balance at basin level and thereby dictates the dryness or wetness of a basin. Early empirical formulations [30,32,33] represented the general hydrological reactions of catchments to global climate forcing, as they were based on continental or global runoff information. In 1981, Fu [34] incorporated the catchment characteristic parameter (parameter n) into catchment water balance. Parameter n comprehensively reflects catchment characteristics, including the characteristics of soil, topography, geologic properties, and vegetation cover [31,35]. In 2008, Yang et al. [36] derived the Choudhury–Yang formula based on Choudhury's [37] proposed empirical theoretical equation. The early consistency of the rich theory and the later improvements to the methodology increased the credibility of the method. Therefore, coupled with the elasticity coefficient method, the Choudhury–Yang method has been successfully employed to analyze and separate the impacts of climate change and human activities on changes in runoff [3,27,38,39].

As the largest tributary of the Yellow River, the Wei River discharges approximately 19.7% of water into the Yellow River every year, which plays a significant role in the management and sustainable development of water resources [40]. It provides ~85% of the water supply for irrigation, domestic, and industrial use, and nourishes more than 22 million people in this basin, making it an essential factor for regional economic development [41]. However, significant runoff reductions have been reported in many studies [40,42]. With the growth in population and development of the economy, the demand for water resources tends to increase, and water scarcity becomes more severe [43,44]. Moreover, the northern part of the Wei River Basin (WRB) is located in the Loess Plateau, which suffers from serious soil erosion. Catchment characteristics and runoff patterns have been significantly affected by the execution of ecological restoration projects [44–46]. Despite the fact that many studies have separated the influence of climate change and human intervention on WRB runoff [24,40,46], different studies have arrived at disparate conclusions due to their varying methods and periods. Moreover, most studies only focused on alterations in runoff of the entire basin or a single hydrological station but ignored the runoff change in smaller catchments. Therefore, we conducted a comprehensive attribution analysis of runoff change at different catchment scales of the WRB from 1970 to 2018 by using the Budyko-based method. The main objectives of this study were to (1) depict the trend of runoff, P, PET, temperature, and parameter n in different catchments and the change for abrupt year; (2) assess the spatial-temporal variability of the runoff elasticity coefficients

to P, PET, and parameter n in different catchments; and (3) quantitatively estimate the changes in runoff attributable to climate variables and land cover and clarify the possible influencing factors at different spatial scales.

## 2. Study Area and Data

### 2.1. Study Area

The WRB (34–38° N; 104–111° E) is located in the lower midstream of the Yellow River, covering an area of approximately $1.35 \times 10^5$ km$^2$ (Figure 1). The Wei River rises in the Niaoshu Mountain of Gansu Province, through three provinces (Gansu, Shaanxi, and Ningxia), and across the Loess Plateau before entering the Yellow River. Its two primary tributaries are the Jing River and Beiluo River, which divide the Wei River Basin into three sub-basins: the Jing River (JR) basin, Beiluo River (BLR) basin, and the upstream of the Wei River (UWR) basin. Traditionally, the region is named the Jing–Luo–Wei Region, and the catchment characteristics of the three tributaries are very different in space. The WRB is situated in the continental monsoon climate zone, and experiences high temperatures and abundant P during summer, whereas winter brings low temperatures and scarce P. The average annual P is approximately 536 mm. The distribution of P is uneven across space, with the majority of it (over 60%) being concentrated within the months from June to September (Figure 2).

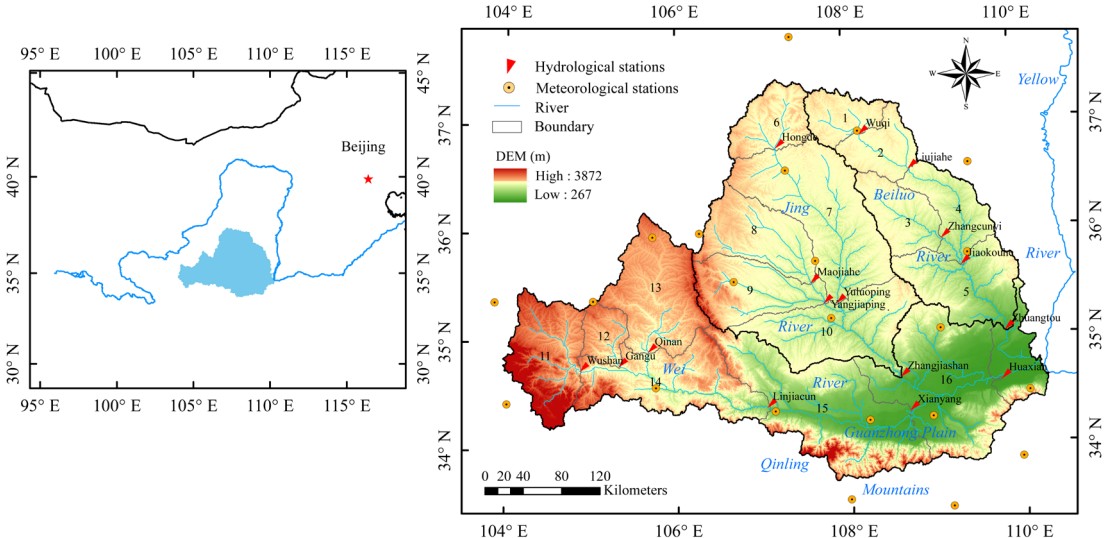

**Figure 1.** Location of the study region and hydrological and meteorological stations in the Wei River Basin.

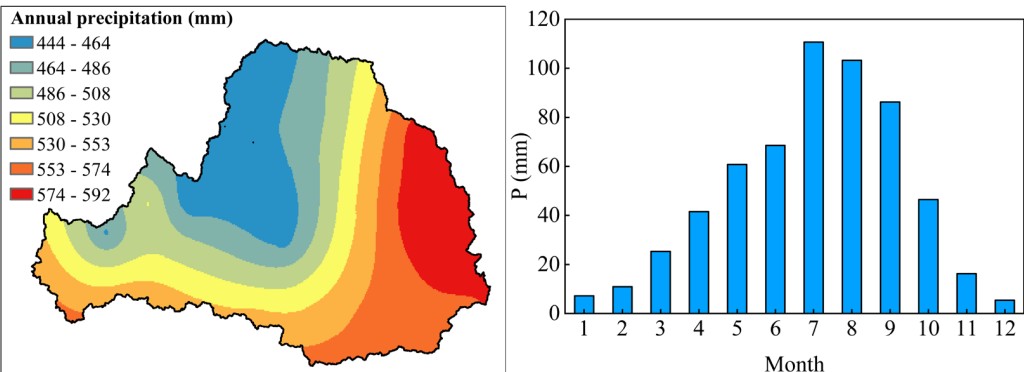

**Figure 2.** Spatial distribution of multi-year average precipitation and intra-annual distribution of precipitation in the Wei River Basin.

*2.2. Data*

Annual runoff data for 16 hydrological stations from 1970 to 2018 were made available by the Yellow River Water Resources Commission. Daily meteorological data (1970–2018) from 22 meteorological stations were collected from the National Meteorological Information Centre of China (NMIC) (https://www.nmic.cn/ (accessed on 1 January 2020)), and consisted of precipitation, wind speed, relative humidity, sunshine duration, and average, maximum, and minimum temperature. The information of the hydrological stations is shown in Table S1 in the Supplementary Materials. Land cover datasets from 1980, 1995, and 2018 were available at the Chinese Academy of Sciences Resource and Environmental Science Data Center (http://www.resdc.cn/ (accessed on 1 January 2020)). The land cover was divided into six classifications: cropland, forest, settlement, grassland, water, and others, with a spatial resolution of 1 km. The 15 day Normalized Difference Vegetation Index (NDVI) dataset was acquired from the Global Inventory Modeling and Mapping Studies (GIMMS) NDVI3g vegetation index dataset designed by NASA (https://ladsweb.modaps.eosdis.nasa.gov/ (accessed on 1 January 2020)), with a spatial resolution of 500 m from 1982 to 2015. The annual mean and maximum NDVI values were used in this study.

The entire basin was divided into 16 catchments, considering the distribution of hydrological stations, to examine the spatial features of runoff change. The catchment controlled by a single hydrological station was divided into one catchment, and the runoff in that catchment was equal to the observed runoff of this hydrological station. For instance, the runoff value of catchment 1 was equal to the runoff observed at the Wuqi station. The catchment controlled by adjacent hydrological stations was divided into one catchment, and the runoff value of this catchment was calculated using the runoff difference between the upstream and downstream hydrological stations. For example, the runoff of catchment 2 was the observed runoff of Liujiahe minus that of Wuqi. Table S2 in the Supplementary Materials presented comprehensive details about the catchments.

## 3. Methodology

### 3.1. Trend Detection

The non-parametric Mann–Kendall (M–K) method was implemented to detect the trends for annual runoff, P, PET, and parameter n in the WRB [47,48]. This method was selected for trend detection in hydro-meteorological records because its robustness demands non-normality and censoring data.

For a given dataset X (i = 1, 2, 3, . . . , n), the test statistic (S) is given by

$$S = \sum_{i=1}^{n-1} \sum_{i+1}^{i} sgn(X_i - X_j) \tag{1}$$

where

$$sgn(x_j - x_i) = \begin{cases} 1 & x_j - x_i > 0 \\ 0 & x_j - x_i = 0 \\ -1 & x_j - x_i < 0 \end{cases} \tag{2}$$

A standardized statistic Z is obtained by

$$Z = \begin{cases} (S-1)/\sqrt{n(n-1)(2n+5)/18} & S > 0 \\ 0 & S = 0 \\ (S+1)/\sqrt{n(n-1)(2n+5)/18} & S < 0 \end{cases} \tag{3}$$

To reduce the influence of the serial correlation, the trend-free pre-whitening test was applied to identify any temporal trends in hydro-meteorological variables [49], which is defined as

$$Y_t = X_t - rX_{t-1} \tag{4}$$

where $Y_t$ is the de-trended and pre-whitened series, referred to as the residual series, and $X_t$ represents the original time series. If the lag-one serial coefficient r of the de-trended series is statistically significant at the 5% level, it means that the serial correlation has been eliminated. Additionally, a Mann–Kendall test was conducted on the whitened series to obtain a Z-statistic. A positive Z-value indicates an upward trend, while a negative value indicates a downward trend [40]. $|Z| < 1.96$ means no significant trend was detected ($p > 5\%$). $|Z| \geq 1.96$ and $|Z| \geq 2.58$ indicate that the significance test has passed with a confidence level of 95% and 99%.

*3.2. Breakpoint Analysis*

In this study [49], Pettitt's test was applied to identify the abrupt year in the hydrological data by utilizing a significance level of 5% through a non-parametric approach [50]. This test detects the occurrence of breakpoints using a non-parametric method at a significance level of 5%. This test applies a function of the Mann–Whitney statistic ($U_{t,n}$), which has been mentioned in the Mann–Kendall test. When a significant breakpoint is detected, it indicates that the two subseries ($x_1, \ldots, x_t$ and $x_{t+1}, x_{t+2}, \ldots, x_n$) are from different populations. $U_{t,n}$ is expressed as follows:

$$U_{t,n} = \sum_{i=1}^{t} \sum_{j=1}^{n} \mathrm{sgn}(X_t - X_j) \quad \text{If } t = 2, \ldots, n \tag{5}$$

The test statistic $K_n$ is defined as maximum value of $|U_{t,n}|$

$$K_n = \max|U_{t,n}| \tag{6}$$

The associated probability (p) with $K_n$ is given by:

$$p = \exp\left(\frac{-6(K_n)^2}{n^3 + n^2}\right) \tag{7}$$

The null hypothesis of Pettitt's test is that there is no change point.

*3.3. Assessing the Impacts of Climatic and Anthropogenic Factors on Runoff Change*
3.3.1. Budyko Framework

For a given catchment, the long-term water balance equation can be written as

$$R = P - AET - \Delta S \tag{8}$$

where R, P, AET, and $\Delta S$ are denoted as the runoff depth (mm), precipitation (mm), actual evapotranspiration (mm), and water storage change, respectively. Here, the variation in water storage can be ignored for the given hydrological year and is assigned a value of zero.

Budyko [30] proposed that the ratio of PET to P over a long-term average can be used to describe the hydrological drought conditions of a region. This basis for the hypothesis is that the water and energy balance processes are regulated by climate and topography, and thus any changes in these factors will lead to corresponding variations in runoff. Over the years, this hypothesis has been widely utilized in hydrology and hydrological modelling to understand the impacts of climate and land surface characteristics on runoff processes [30,36,37]. To separate the influence on runoff independently, this study adopted the elasticity coefficient method coupled with the Budyko frameworks. AET can be derived from the Choudhury–Yang equation:

$$AET = \frac{P \times PET}{(P^n + PET^n)^{1/n}} \tag{9}$$

where PET represents the potential evapotranspiration, and parameter n comprehensively reflects the catchment characteristics, including the characteristics of soil, topography, geologic properties, and vegetation cover [35].

### 3.3.2. Elasticity Coefficient

Combining Equations (8) and (9), we obtain an equation R = $f$ (P, PET, n). P is the observed annual mean precipitation. PET can be estimated using the FAO Penman–Monteith [51], which is currently recognized as an effective method for estimating PET. The method takes into account variations in meteorological factors such as temperature, humidity, wind speed, and solar radiation, as well as the complexity of vegetation cover, and is used to estimate PET in different regions/basins to help design irrigation systems and improve crop management. The parameter n can be derived from this equation using MATLAB R2022b software.

The total differential equation of R with different factors can be written as

$$dR = \frac{\partial f}{\partial P}dP + \frac{\partial f}{\partial PET}dPET + \frac{\partial f}{\partial n}dn \tag{10}$$

The elasticity of runoff to a specific independent variable change is the degree of the response of runoff to the changes in variables, e.g., P, PET, or n, which can be estimated by the proportional change in simulated runoff. The elasticity of runoff is given as

$$\varepsilon_{x_i} = \frac{\partial R}{\partial x_i} \times \frac{x_i}{R} \tag{11}$$

where $\varepsilon_{x_i}$ is the elasticity of runoff to P, PET, or n. Therefore, Equation (10) is written as

$$dR = \varepsilon_{x_P}\frac{R}{x_P}dx_P + \varepsilon_{x_{PET}}\frac{R}{x_{PET}}dx_{PET} + \varepsilon_{x_n}\frac{R}{x_n}dx_n \tag{12}$$

The elasticity of runoff is expressed as [3]:

$$\varepsilon_P = \frac{1 - \left[\frac{(PET/P)^n}{1+(PET/P)^n}\right]^{1/n+1}}{1 - \left[\frac{(PET/P)^n}{1+(PET/P)^n}\right]^{1/n}} \tag{13}$$

$$\varepsilon_{PET} = \frac{1}{1 + (PET/P)^n}\frac{1}{1 - \left[\frac{1+(PET/P)^n}{(PET/P)^n}\right]^{1/n}} \tag{14}$$

$$\varepsilon_n = \frac{A - B}{\left[1 + (PET/P)^n\right]^{1/n} - 1} \tag{15}$$

$$A = \frac{P^n\ln(P) + PET\ln(PET)}{P^n + PET} \tag{16}$$

$$B = \frac{\ln(P^n + PET^n)}{n} \tag{17}$$

### 3.3.3. Attribution Analysis of Runoff Changes

According to the breakpoint, the whole time series can be divided into two periods: "period 1" which represents the base period before the abrupt year, and "period 2" which represents the human impact period after the abrupt year. Thus, the change in annual runoff from period 1 to period 2 is given by

$$\Delta R = R_2 - R_1 \tag{18}$$

where $\Delta R$ is the difference between the mean annual observed runoff in the two different periods, $R_1$ is the average runoff in the base period, and $R_2$ is the average runoff in the human impact period.

The total runoff change over a long period can be estimated as follows:

$$\Delta R_{total} = \Delta R_{clima} + \Delta R_{human} \tag{19}$$

where $\Delta R_{clima}$ is the change in runoff depth due to climatic variables, including the change in runoff caused by P ($\Delta R_P$) and PET ($\Delta R_{PET}$). $\Delta R_{human}$ is the runoff change caused by human activities, equal to the change in runoff caused by the catchment characteristics of parameter n. The simulated runoff change can be calculated as follows:

$$\Delta R_P = \varepsilon_P \frac{R}{P} \Delta P$$
$$\Delta R_{PET} = \varepsilon_{PET} \frac{R}{PET} \Delta PET \tag{20}$$
$$\Delta R_n = \varepsilon_n \frac{R}{n} \Delta n$$

$\Delta P$, $\Delta PET$, and $\Delta n$ represent the difference in P, PET, and parameter n from period 1 to period 2.

The relative contribution is the percentage of P, PET, and parameter n to the runoff change, which is expressed as

$$C_P = \frac{\Delta R_P}{\Delta R} \times 100\%$$
$$C_{PET} = \frac{\Delta R_{PET}}{\Delta R} \times 100\% \tag{21}$$
$$C_n = \frac{\Delta R_n}{\Delta R} \times 100\%$$

where $C_P$, $C_{PET}$, and $C_n$ are the contribution proportions of climate variation (P and PET) and anthropogenic activities to runoff change, respectively.

The performance of the Budyko-based method can be evaluated by Nash–Sutcliffe efficiency values (NSE) and the absolute values of the relative error ($R_e$).

$$NSE = 1 - \frac{\sum_{i=1}^{n}(\Delta R_1 - \Delta R_2)^2}{\sum_{i=1}^{n}(\Delta R_1 - \Delta R_{1a})^2} \tag{22}$$

$$R_e = \frac{\Delta R_1 - \Delta R_2}{\Delta R_1} \times 100\% \tag{23}$$

$\Delta R_1$ is the observed runoff change, $\Delta R_2$ is the runoff change estimated by Budyko-based method. $\Delta R_{1a}$ is the average observed runoff change.

## 4. Results

### 4.1. Trends in Hydro-Meteorological Factors

To better understand the variation in runoff and its influencing factors at different scales, we analyzed the trends in the annual runoff, P, PET, temperature, and parameter n within the three sub-basins (UWR, JR, and BLR) of the WRB from 1970 to 2018 (Figure 3). Significantly decreasing trends were observed for runoff ($p < 0.01$), with average rates of 0.87 mm/y, 0.60 mm/y, and 0.19 mm/y in the UWR, JR, and BLR, respectively. P presented downward trends in the UWR and BLR and an upward trend in the BLR, though all trends were found to be non-significant. PET in the UWR and JR showed upward trends (no significance at $p > 0.05$), and both increased by 1.15 mm/y, while the PET of the BLR presented downward trends. Temperature showed a significant increase trend ($p < 0.01$) in all three regions with average rates of 0.03 °C/y, 0.04 °C/y, and 0.02 °C/y in the UWR, JR, and BLR, respectively. In terms of parameter n, the UWR and JR displayed a significant increasing trend ($p < 0.01$), with an average rate of 0.01 mm/y and 0.02 mm/y, respectively, while no significant trend was found for parameter n in the BLR.

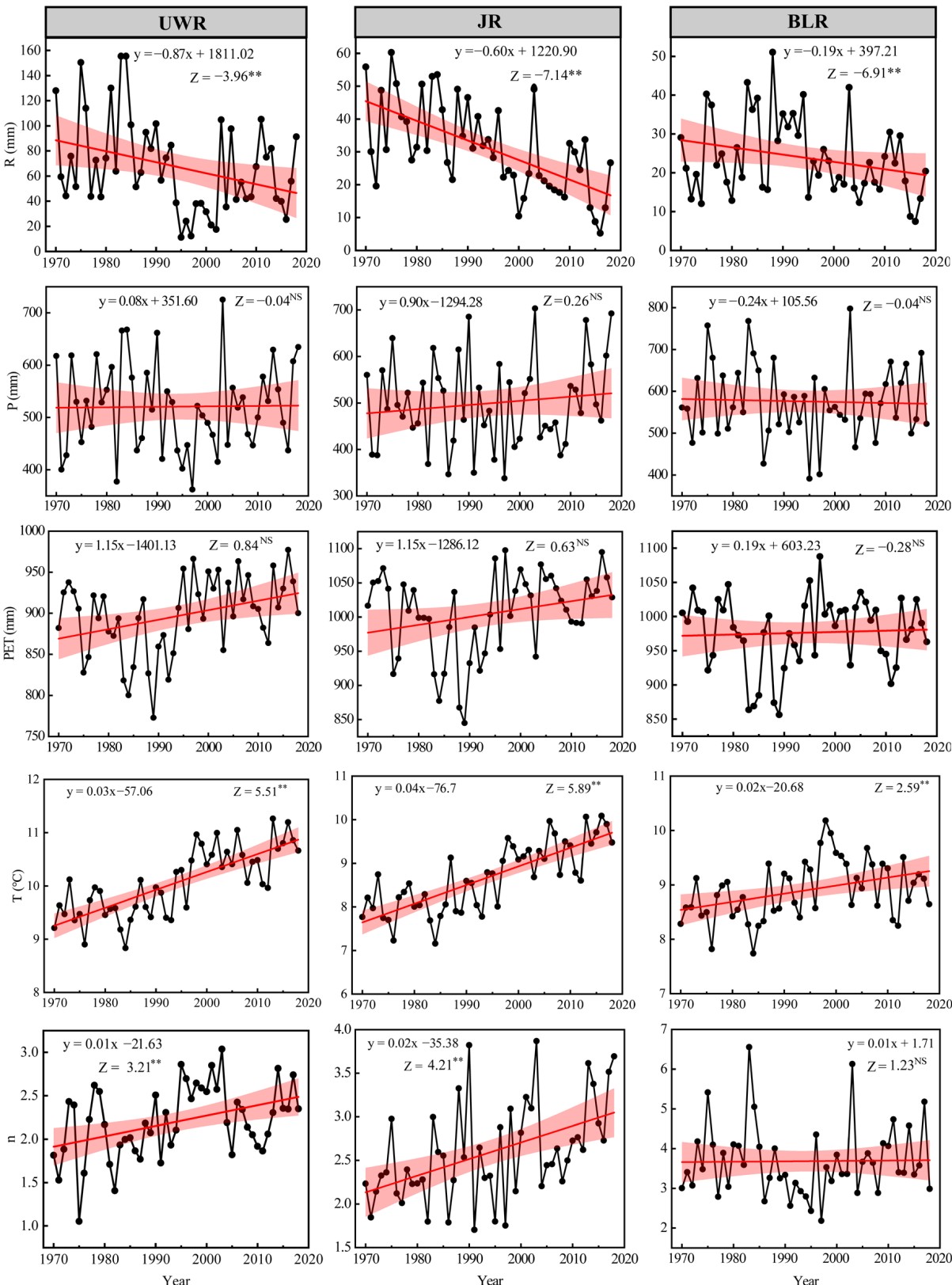

**Figure 3.** Observed annual runoff depth (R); precipitation (P); potential evapotranspiration (PET); temperature (T) and parameter n in the upstream of the Wei River (UWR), Jing River (JR), and Beiluo River (BLR). The solid red lines indicate the trend; the Z value was detected using the Mann–Kendall test (**: significant at $p < 0.01$; NS: no significance at $p > 0.05$). The 95% confidence interval is denoted by the pink strips.

The trends in annual runoff, P, PET, and parameter n from 1970 to 2018 are shown in Figure 4. Out of all the catchments, 12 catchments (which make up 75% of the catchments) exhibited a notable decrease trend ($p < 0.05$) in annual runoff. P showed an increase trend in most catchments, but all of them were statistically insignificant. PET in half of the catchments showed significant increasing trends. A significant increase trend ($p < 0.05$) in temperature was observed in 15 catchments (accounting for 94% of catchments). In addition, parameter n showed a significant increase trend ($p < 0.05$) in 75% of the catchments. The significant increase trend of parameter n indicates that the disturbance of catchment characteristics is gradually intensified.

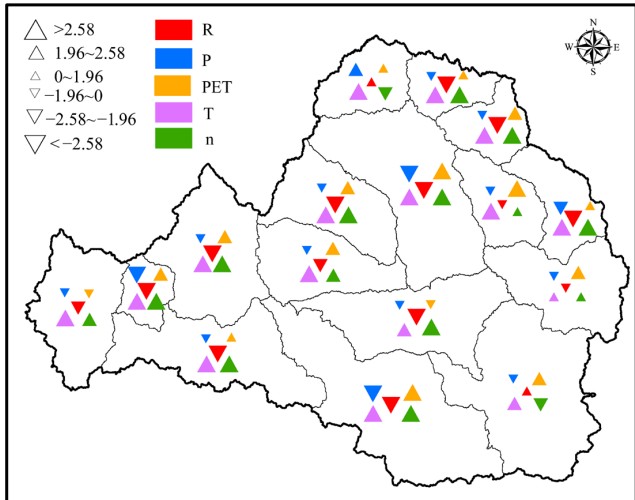

**Figure 4.** Trend analysis of hydro-meteorological variables in 16 catchments of the Wei River Basin. Trends in annual runoff depth (R); precipitation (P); potential evapotranspiration (PET); temperature (T); and parameter n. △ and ▽ denote increasing and decreasing trends, respectively.

### 4.2. Abrupt Years of Runoff Change

The abrupt years of annual runoff were determined using Pettitt's method in the WRB and three sub-basins (UWR, JR, and BLR) from 1970 to 2018 (Table 1). In the WRB, the runoff underwent an abrupt change in 1993 ($p < 0.05$). In the three sub-basins, significant abrupt years ($p < 0.01$) were detected in the UWR (1993), JR (1996), and BLR (1994). It was found that the change in runoff and P decreased after the abrupt year, while that in PET and parameter n increased in the WRB and the three sub-basins. 62.5% of the catchments exhibited significant abrupt changes ($p < 0.05$) around the early 1990s and around 2000. The change in hydro-meteorological characteristics and the contribution of climatic and anthropogenic factors to runoff changes in 16 catchments are shown in Table S3 in the Supplementary Materials.

**Table 1.** The change in the hydro-meteorological characteristics and the elasticity coefficients of runoff to precipitation (P), potential evapotranspiration (PET), and parameter.

| Basin | Abrupt Year | Period | Change before and after the Abrupt Year | | | | Elasticity Coefficient | | |
|---|---|---|---|---|---|---|---|---|---|
| | | | R (mm) | P (mm) | PET (mm) | n | $\varepsilon_P$ | $\varepsilon_{PET}$ | $\varepsilon_n$ |
| WRB | 1993 * | 1970–1993 | −12.3 | −3.2 | 28.7 | 0.4 | 3.15 | −2.15 | −2.12 |
| | | 1994–2018 | | | | | 3.28 | −2.28 | −2.38 |
| UWR | 1993 ** | 1970–1993 | −15.3 | 11.2 | 55.7 | 0.5 | 2.57 | −1.57 | −1.69 |
| | | 1994–2018 | | | | | 3.07 | −2.07 | −2.19 |
| JR | 1996 ** | 1970–1996 | −18.3 | −1.1 | 52.5 | 0.5 | 3.05 | −2.05 | −2.37 |
| | | 1997–2018 | | | | | 3.64 | −2.64 | −2.94 |
| BLR | 1994 ** | 1970–1994 | −24.0 | −19.0 | 44.8 | 0.3 | 3.31 | −2.31 | −2.41 |
| | | 1995–2018 | | | | | 3.34 | −2.34 | −2.46 |

Notes: abrupt year: **: significant at $p < 0.01$; *: significant at $p < 0.05$.

### 4.3. The Elasticity Coefficient of Runoff to Climate Change and Anthropogenic Factors

Table 1 presents the elasticity coefficients of runoff with respect to P, PET, and parameter n for the WRB and the three sub-basins. Compared to the base period, there was a noticeable increase in the absolute values of the runoff elasticity coefficient compared to the base period, implying an increasing sensitivity of runoff change to climate variables and parameter n. For the WRB, the elasticity coefficients after the abrupt year were 3.28 for P, –2.28 for PET, and –2.38 for parameter n. These values suggest that a 10% reduction in P would cause a 32.8% decline in runoff, whereas a corresponding decrease in PET or parameter n would result in an increase of 22.8% and 23.8%, respectively. The results indicated that the highest and positive response of runoff was to P, while it exhibited a negative response to PET and parameter n.

The elasticity coefficients of runoff in 16 catchments concerning P, PET, and parameter n are shown in Figure 5. The spatial distribution patterns of the runoff elasticity coefficient in response to P exhibit similarities with those of PET, but are different in polarity. The elasticity coefficient of runoff to P ranged from 2.6 to 4.2 (Figure 5a). Contrary to P, the elasticity coefficients (–3.2 to –1.6) of runoff were negative for PET (Figure 5b). In the middle and downstream regions of the BLR, the absolute elasticity coefficients of runoff with respect to precipitation (P) and potential evapotranspiration (PET) were found to be higher compared to other regions across the WRB. This suggests that changes in P and PET have a greater impact on the amount of runoff in these regions. The elasticity coefficients of runoff to parameter n ranged from –3.4 to –1.6, and higher absolute values were found in the JR and BLR (Figure 5c). This finding suggested that alterations in catchment characteristics have a greater impact on the runoff variations in these specific regions as compared to others.

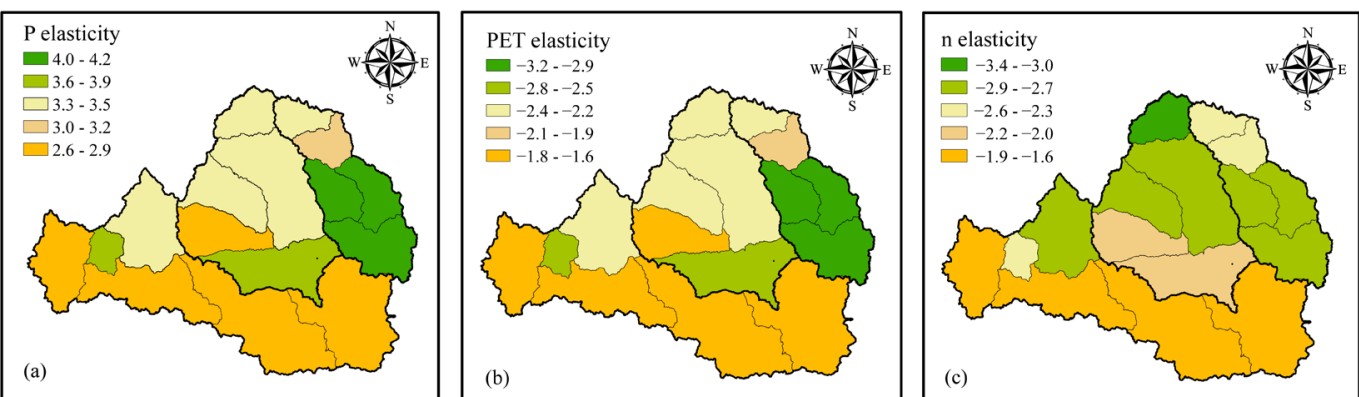

**Figure 5.** The elasticity of runoff to precipitation (**a**), elasticity of runoff to potential evaporation (**b**), elasticity of runoff to parameter catchment characteristics of parameters (**c**) in 16 catchments of the Wei River Basin. The elasticity of runoff means the percent change in runoff coming from the change in precipitation/potential evaporation/catchment characteristics of parameters by 1%.

### 4.4. Evaluating the Impacts of Climatic Anthropogenic Factors on Runoff

The performance of the Budyko-based method is shown in Figure 6. Upon comparing the observed and simulated changes in runoff, it was found that the regression slopes of both close to 1. The NSE values were greater than 0.99, and the absolute values of the $R_e$ were less than 5%. This indicates that the method has a strong modeling capability for runoff changes in most catchments.

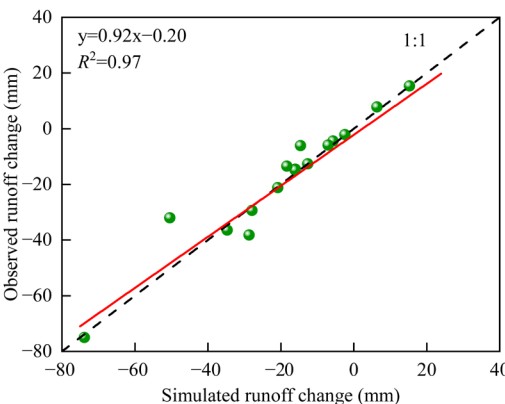

**Figure 6.** Comparison of the observed and simulated runoff change. The solid red lines mean the regression line of the observed and simulated runoff, and the black dotted line represents the regression line with a slope of 1.

Table 2 demonstrates the runoff alterations caused by hydro-meteorological variables (P and PET) and the catchment characteristics parameter n in the WRB and the three sub-basins. The findings indicate that the decreased runoff, induced by P, PET, and parameter n in the WRB and the three sub-basins, may be affected by the decrease in P and the increase in PET and parameter n. The values obtained for parameter n are related to integrated catchment characteristics, including vegetation cover, soil properties, and topography [52,53]. Regarding soil properties and topography as static variables, land cover change is an important factor affecting catchment hydrological behavior [3]. Thus, the impact of various human activities such as afforestation, deforestation, soil and water conservation projects, and surface water consumption will ultimately be indicated by parameter n. The study found that climate variability, which includes P and PET, contributed to 37.2% of the changes in runoff, while human activities accounted for 62.8%. Furthermore, the impact of human activities on the decrease in runoff was particularly significant in each sub-basin, with a contribution rate exceeding 50%.

**Table 2.** The runoff change in hydro-meteorological characteristics and the contribution of climate change and human activities to runoff changes in the WRB and three sub-basins.

| Region | Runoff Change Induced by P/PET/n (mm) | | | Contribution Rate to Runoff Change (%) | | | Contribution Rate of Climate Change (%) | Contribution Rate of Human Activities (%) |
|---|---|---|---|---|---|---|---|---|
| | $\Delta R_P$ | $\Delta R_{PET}$ | $\Delta R_n$ | $C_P$ | $C_{PET}$ | $C_n$ | | |
| WRB | −6.0 | −5.0 | −18.6 | 20.3 | 16.9 | 62.8 | 37.2 | 62.8 |
| UWR | −11.2 | −9.3 | −21.2 | 26.8 | 22.3 | 50.9 | 49.1 | 50.9 |
| JR | −0.2 | −3.4 | −15.3 | 1.1 | 17.9 | 80.9 | 20.8 | 79.2 |
| BLR | −3.4 | −3.2 | −9.7 | 20.7 | 19.5 | 59.8 | 40.2 | 59.8 |

$\Delta R_P$, $\Delta R_{PET}$, and $\Delta R_n$ are the runoff changes induced by the change in P, PET, parameter n after the abrupt year, respectively. $C_P$, $C_{PET}$, and $C_n$ are the relative contributions of P, PET, parameter n to runoff change.

It can be observed from Figure 7 that the runoff changes before and after the abrupt year induced by hydro-meteorological variables and parameter n is spatially inconsistent. The change in P positively impacted runoff (runoff increase) in most catchments, while changes in PET and parameter n had a negative influence on runoff (runoff decrease) in all catchments. The runoff change induced by P, PET, and parameter n between two periods ranged from −8.5 to 15.6 mm, −7.5 to −1.2 mm, and −38.1 to 12.5 mm, respectively. Figure 8 shows the contribution proportions of climate variables (P and PET) and catchment characteristics (parameter n) to runoff change. Except for catchments 3 and 4, all catchments showed that catchment characteristic changes induced by human activities contributed more than climate variables to runoff change. Therefore, catchment characteristic changes

were the dominant factors shaping runoff in most catchments, while climate change contributed the most to runoff in two catchments (3 and 4) in the middle of the BLR. For sub-region 3, the increased PET (38.5 mm) from the base period to the human impact period had the most significant impact on runoff reduction, accounting for 113%. For catchment 4, P (52.2 mm) increased by 52 mm after the abrupt year, and the runoff change induced by P was 8.3 mm, which accounted for 107%.

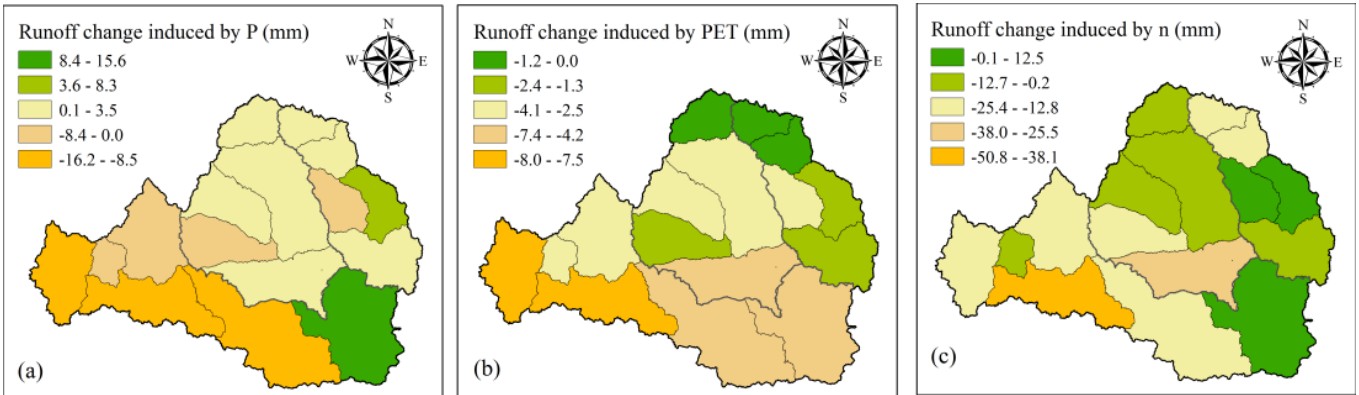

**Figure 7.** Runoff change induced by P (**a**), PET (**b**) and parameter n (**c**) in 16 catchments of the Wei River Basin. P: precipitation; PET: potential evapotranspiration; n: catchment characteristic parameter.

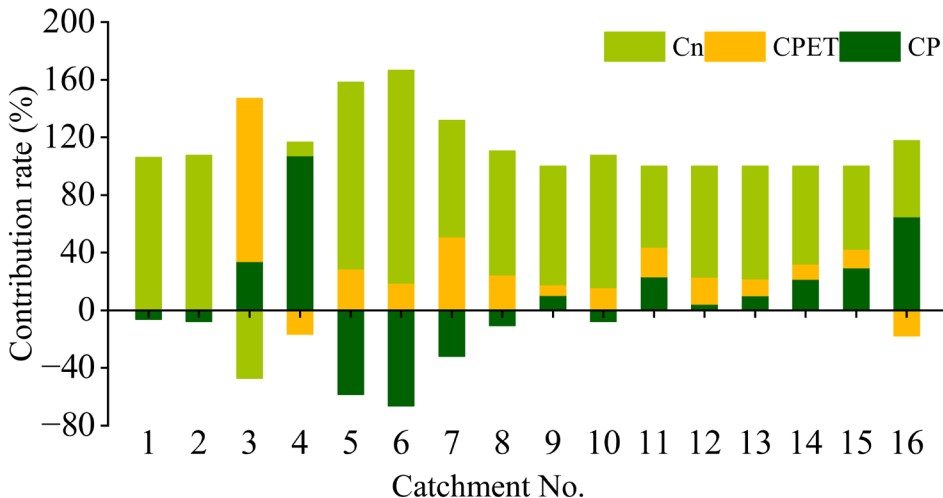

**Figure 8.** Attribution of the annual runoff change in the 16 catchments in the Wei River Basin.

## 5. Discussion

### 5.1. Contribution of the Climatic Factors to Runoff Change

Water resource systems are dramatically influenced by climatic variation, especially in arid and semi-arid areas [7,54]. Previous studies have reported that variations in temperature, P, and PET are manifestations of climate change, which has a significant impact on runoff in the WRB [55,56]. It has been shown that P and temperature are positively associated with terrestrial water storage, and that decreased terrestrial water storage is driven by decreased P and increased PET [57]. Our study indicates that the temperature of the WRB has shown a notable rise. Pang et al. [58] observed that the temperature in the WRB has increased by 0.42 °C/10a. The high temperature accelerates evapotranspiration, reducing the magnitude of water entering the river channel and peak runoff [38]. The drastic runoff reduction caused by increasing PET in the WRB was also observed by Zhao et al. [55]. However, as the main source of runoff, our study showed that the annual P declined 12.3 mm after 1993, but there was no noticeable decreasing trend in the WRB from 1970 to 2018. Similar decreasing trends in P were also detected in other studies in the

WRB [46,59]. Our results indicated that P showed the highest and most positive correlation with runoff change, but the change in P (20.3%) contributing to the decrease in runoff was slightly greater than that for PET (16.9%) in this region. This is in agreement with the previous results, which identified that changes in PET have a weaker impact on runoff change than those in P [52]. In addition, climate change has led to the reduction in runoff in the WRB from 1970 to 2018, although its impact is far less than that of human activities. In contrast, Luo et al. [60] demonstrated that climate change has increased the annual runoff in the source area of the Yangtze River, which has offset a large part of the annual runoff reduced by the ecological protection plan. This may be attributed to the decrease in P and the increase in PET in the WRB.

*5.2. Contribution of the Catchment Characteristics' Change to Runoff Change and Potential Anthropogenic Activities*

To investigate runoff changes caused by human activities in the WRB, we evaluated three satellite images that recorded land cover. Croplands, forests, and grasslands constituted the major types of land cover in the WRB. Our results indicate the change point in runoff regime in the WRB around 1995, leading to this study focusing on land cover change after the transition (from 1995 to 2018). During this period, the main land cover conversion in the WRB was concentrated from cropland to grassland, settlement, and forest; from grassland to cropland and forest; and from forest to grassland, with a total conversion area of 54,851 km$^2$ (Figure 9a). Consequently, we selected the six main conversion types and computed the proportion of each land conversion type in the overall conversion area based on the total conversion area (Figure 9b). Conversion between cropland and grassland had the highest percentage (around 30%) followed by conversion between forest and cropland (about 11%), while transformations of some cropland to settlement and forest were also presented, which account for 6.4% and 6.5%. Meanwhile, the net change in four main land cover types underwent greater changes from 1995 to 2018 were analyzed: cropland, settlement, forest, and grassland. During this period, cropland decreased by 4203 km$^2$, while forest and grassland increased by 1162 km$^2$ and 1114 km$^2$, respectively. These changes can be partly attributed to large-scale ecological conservation programs, such as the Grain for Green (GFG) projects implemented in 1999, which led to the reconversion of substantial areas of cropland to forest and grassland, substantially increasing the vegetation cover in the WRB [61]. Chen et al. [62] also observed a dramatic change in land cover owing to an increase in vegetation restoration. Moreover, we observed a significant increase in settlement area, with an increase of 2012 km$^2$, which may be related to the urbanization of the local area.

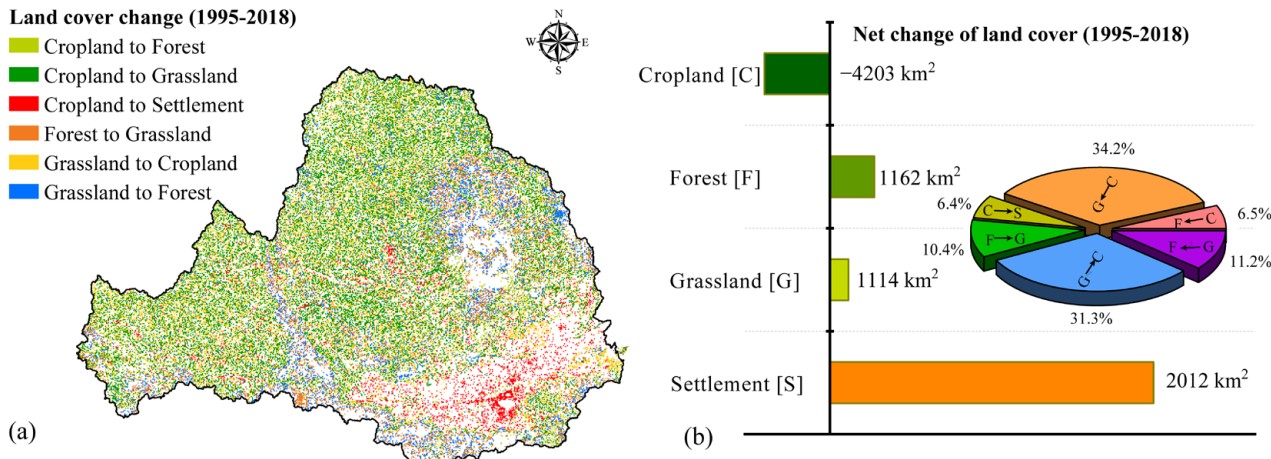

**Figure 9.** Spatial and temporal variability and net change of land cover in the Wei River Basin from 1995 to 2018. (**a**) Land cover change (1995–2018). (**b**) Net change of land cover.

Given the execution of large-scale ecological conservation programs, we assessed vegetation coverage variations using the annual mean NDVI and annual maximum NDVI of the WRB from 1982 to 2015 (Figure 10). The results show that vegetation cover increased significantly ($p < 0.05$) after 2000, which may be attributed to the GFG project executed after 1999. Research has suggested that large-scale vegetation restoration will locally decrease the annual mean water availability and runoff [63,64]. Increased vegetation cover allows more precipitation to be intercepted by the vegetation canopy and root system [65], thereby diminishing terrestrial runoff [66]. Yang et al. [67] found that increased vegetation coverage was the main reason for runoff reduction in the middle reaches of the Yellow River Basin, which decreased runoff by 35% from 2000 to 2005. Similar results were obtained by Liu et al. [68] and Zuo et al. [42], who concluded that vegetation coverage was the dominant factor responsible for runoff reduction. In addition, large-scale soil and water conservation measures (e.g., afforestation, terraces, fish-scale pits, and grazing ban) have played a vital role in vegetation restoration in the WRB. Since the implementation of these measures in the 1980s, runoff has declined by approximately $9.9 \times 10^8$ m$^3$/year (15.5% of runoff at Huaxian station) in the WRB [55]. Moreover, interesting results were observed showing that 62.5% of the catchments have significant abrupt changes ($p < 0.05$), which may attribute to the soil and water conservations measures implemented after the 1990s and Grain for Green project since 2000 [9]. However, for the runoff regime in the WRB and three sub-basins, there was no significant abrupt change detected around 2000. It may be that vegetation restoration is a slow and long-term process, and the hydrological effect of vegetation on runoff in large catchment has a hysteresis effect [69]. Zhao et al. [70] also found that the response of runoff change to vegetation cover change in large catchment (>1000 km$^2$) weakened with the increase in catchment scale.

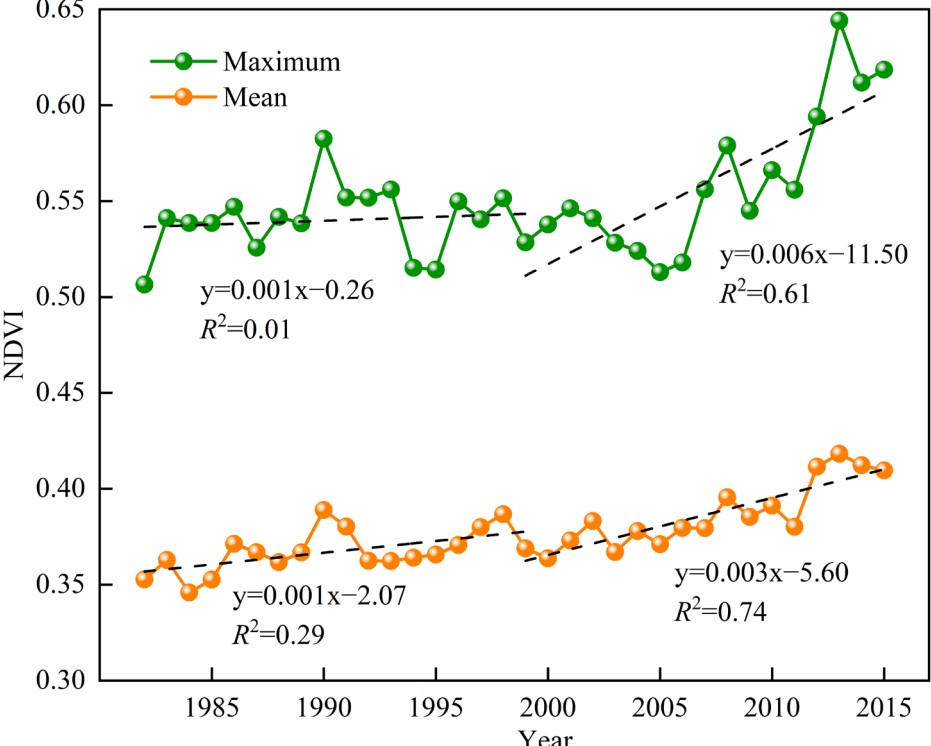

**Figure 10.** The variations in the mean annual NDVI and maximum NDVI in the Wei River Basin from 1982 to 2015.

The impacts of human activities on catchment characteristics are complex, especially in the context of the extensive implementation of ecological conservation programs. In addition to these ecological conservation programs, the WRB has also witnessed many other intensive human activities in the past 30 years, such as urbanization, and large-scale

infrastructure construction (water infrastructures). A large amount of cropland has been converted into settlement (3503 km$^2$) from 1995 to 2018; this change mainly occurred in the Guanzhong Plain owing to urbanization. Intensified urbanization would have also resulted in changes in catchment characteristics by increasing impervious areas, consequently leading to increases in runoff and decreases in infiltration [22]. Additionally, the reservoir/dams construction has a direct impact on the catchment characteristics of parameter n, mainly by affecting the vegetation coverage in the irrigation area to affect the runoff [38]. Our results showed that human activities play a dominant role in runoff decreasing, which is consistent with the findings in other catchments such as Loess Plateau [22], Yellow River [71], and Yangtze River [60].

### 5.3. Limitations of the Study

This study has some notable limitations. Equation (10) is a first-order approximation, which may lead to errors in the calculated impacts of climatic variations and anthropogenic activities. In addition, this framework of contribution estimation supposes that anthropogenic activities and climatic variation are completely independent, but it is presently difficult to easily separate and decompose them [71]. Both climate change and human activities will affect the change in catchment characteristic parameter n. Attributing land cover change to human activities will overestimate the impact of local human activities on runoff change [3].

## 6. Conclusions

This study quantified the impact of climate change and human-induced land cover variations on runoff change in the WRB using the elasticity coefficient method coupled with decomposition methods based on the Budyko framework. The results show the following:

(1) The annual runoff and P presented a statistically downward trend and a significant decrease in runoff was detected in all three sub-basins ($p < 0.01$) and 75% of the catchments ($p < 0.05$). However, PET and parameter n showed an increasing trend in the WRB, three sub-basins, and most catchments.

(2) The significant abrupt years of runoff change were observed in the mid-1990s at the WRB and the three sub-basins ($p < 0.01$); 62.5% of the catchments exhibited significant abrupt changes ($p < 0.05$) in the early 1990s and around 2000.

(3) The results of the elasticity coefficient revealed that P had the highest positive correlation with runoff change, but PET and parameter n had a negative correlation. During the human impact period, a 10% decrease in P would result in a 32.8% drop in runoff, whereas a 10% decrease in PET and parameter n would induce a 22.8% and 23.8% increase in runoff, respectively.

(4) Land cover change induced by human activities had a greater impact on runoff decline than climatic factors in the entire basin, three sub-basins, and most catchments. The impacts of P, PET, and parameter n resulted in average runoff reductions of 20.3, 16.9, and 62.8%, respectively, in the WRB.

These findings highlight the importance of systematically investigating the impacts of climate change and land cover on runoff reduction using the Budyko-based method at different catchment scales. This study could contribute to better quantitative clarification of the driving factors of runoff variation and provide a reference foundation for sustainable water management and ecosystem health conservation across the WRB and midstream of the Yellow River. However, considering the complex interactions among diverse human activities, accurately separating the contributions of other human activities remains challenging. More human activity metrics (e.g., GDP and population) should be considered when interpreting runoff changes, and more methods should be applied to verify the accuracy of the results in future studies.

**Supplementary Materials:** The following supporting information can be downloaded at: https://www.mdpi.com/article/10.3390/w15112081/s1, Table S1: Information on the hydrological stations;

Table S2: Information on the hydro-meteorological characteristics of the 16 sub-regions in the Wei River Basin; Table S3: The change in hydro-meteorological characteristics and contribution of climatic and human activity factors to runoff changes in catchments.

**Author Contributions:** Conceptualization, R.X., X.M. and P.G.; Methodology, R.X., C.G., D.Q. and C.W.; Resources, C.G., D.Q., C.W., X.M. and P.G.; Writing—original draft, R.X.; Writing—review & editing, D.Q. and P.G.; Supervision, X.M.; Funding acquisition, P.G. All authors have read and agreed to the published version of the manuscript.

**Funding:** This research was funded by National Key Research and Development Program of China, Grant/Award Number: 2016YFC0501707.

**Data Availability Statement:** The daily series of meteorological data used in this study are available on National Meteorological Information Centre of China (NMIC) (https://www.nmic.cn/), while the data sharing of runoff are not applicable.

**Conflicts of Interest:** The authors declare no conflict of interest.

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
