# Peer review of "Analysis of Runoff Changes in the Wei River Basin, China: Confronting Climate Change and Human Activities"

_water, doi:10.3390/w15112081_

Round 1
Reviewer 1 Report
Review Report on the Manuscript Number: water-2343364 Title: Analysis of Runoff Changes in the Wei River Basin, China: Confronting Climate Change and Human Activities
I have finished my review on the proposed paper " Analysis of Runoff Changes in the Wei River Basin, China: Confronting Climate Change and Human Activities ".
1. Generally, the manuscript presents an interesting topic and the specific research seems to include some significant points for the research community of this field.
2. The proposed paper is well written with very good use of English language. Except some very minor grammatical mistakes and word errors. The authors should check again the paper to correct these minor mistakes.
3. The proposed paper is very well structured. It begins with an analytical Introduction with the appropriate references that helps the reader to get into the subject immediately. In Introduction there is an effort to provide previous studies with similar scientific content, which took place in the research area and in other countries. Authors describe and set very well the scientific problem and how other researchers have approached. At the end of Introduction, authors clearly state the goals of the research.
The Abstract is a little confusing for readers. The methods used should be presented completely.
Line 27-28, It would be good to add some works about the effects of climate variability and human activities on runoff. Please read and add references as follows:
Sidong Zeng, Chesheng Zhan, Fubao Sun, Hong Du, and Feiyu Wang. Effects of Climate Change and Human Activities on Surface Runoff in the Luan River Basin. 2015.
Misagh Parhizkar , Mahmood Shabanpour, Manuel Esteban Lucas-Borja , Demetrio Antonio Zema, Siyue Li, Nobuaki Tanaka, Artemi Cerda. Effects of length and application rate of rice straw mulch on surface runoff and soil loss under laboratory simulated rainfall. International Journal of Sediment Research 36 (2021) 468e478.
Meilin Wang,Yaqi Shao,Qun’ou Jiang…. Impacts of Climate Change and Human Activity on the Runoff Changes in the Guishui River Basin. Land 2020, 9(9), 291.
4. The methodology is generally very interesting, and well explained, so other researchers could easily repeat it.
5. The results are OK.
6. The Discussion is very qualitatively.
Please explain relation and extension of results obtained from the study to natural conditions in larger scales (scaling).
7. Conclusions are appropriate for this paper.
Reviewer 2 Report
Observational evidence demonstrated that hydrological cycle throughout the world has been dramatically affected by climate change and human activities. As a key component of hydrological cycle, runoff performs a crucial role in water resources management. Its variability has an important implications on water usage patterns in different sectors such as agriculture, industry, households, hydropower generation, and navigation. However, at different spatial scales, the response of runoff to climate change and human activities varies. This was conducted in the WRB including three sub-basins and sixteen catments to detect the long-term (1970-2018) trends in hydro-meteorological variables and quantify the contribution of climate change and anthropogenic factors on runoff change. The results show that Anthropogenic activities had a greater impact on runoff reduction in the WRB (62.8%), three sub-basins (53.9% to 65.8%), and most catchments (–47.0% to 147.3%) than climate change. Dramatic catchment characteristics changes caused by large-scale human activities were the predominant reason of runoff reduction in the WRB.
Overall, it is a nice work and it fit well with the scope of Water and the topic is generally interesting to hydrology scientists. The manuscript is well-written, with clear and concise presentation of the results, accompanied by informative and visually appealing figures. I am pleased to recommend its publication in the Water. Some minor suggestions or comments listed as follows (mostly in formats or spelling), and I hope these comments would be useful for improving the quality of manuscript.
1. There is a small typo in line 81, please delete one full stop after scales.
2. In different sentences, the use of hyphen is not uniform, such as line 158 and line 132, and the author is expected to use uniform and correct punctuation mark.
3. Line 201, there is one minor issue: "thoughall" should be two separate words, "though all."
4. In line 240, the sentence "… which implying an increasing sensitivity of runoff change to climate var-iables and parameter n", "which implying" should be "implying", and it should read: "Compared to the base period, all absolute values of the runoff elasticity coefficient showed an increasing trend after the abrupt year, implying an increasing sensitivity of runoff change to climate variables and parameter n."
5. In line 273, the sentence "The values obtained for parameter n is related to integrated catchment characteristics", "is" should be "are" because "values" is plural.
6. Line 331, "each land conversion types" should be "each land conversion type"
7. Line 347, "may attribute" should be "may be attributed"
8. Line 357, "an interesting results were observed" should be "interesting results were observed"
Reviewer 3 Report
The conclusion should be summarized on the results and discussion. So I suggest to modify the conclusion part like this. I'd like you to analyse furtherly the contribution of each type of human actions and their spatial heterogeneity.
Reviewer 4 Report
I am not an hydrologist, but the paper seems quite solid
